# Stability Analysis of the Explicit Difference Scheme for Richards Equation

**DOI:** 10.3390/e22030352

**Published:** 2020-03-18

**Authors:** Fengnan Liu, Yasuhide Fukumoto, Xiaopeng Zhao

**Affiliations:** 1School of Mathematical and Physical Sciences, Dalian University of Technology, Panjin 124221, China; liufengnan@dlut.edu.cn; 2Institute of Mathematics for Industry, Kyushu University, 744 Motooka, Nishi-ku, Fukuoka 819-0395, Japan; 3College of Sciences, Northeastern University, Shenyang 110819, China; zhaoxiaopeng@jiangnan.edu.cn

**Keywords:** Richards equation, explicit difference scheme, stability analysis

## Abstract

A stable explicit difference scheme, which is based on forward Euler format, is proposed for the Richards equation. To avoid the degeneracy of the Richards equation, we add a perturbation to the functional coefficient of the parabolic term. In addition, we introduce an extra term in the difference scheme which is used to relax the time step restriction for improving the stability condition. With the augmented terms, we prove the stability using the induction method. Numerical experiments show the validity and the accuracy of the scheme, along with its efficiency.

## 1. Introduction

A knowledge of the way of infiltration of water into the ground is crucially important for predicting disasters, such as river floods and landslides, when heavy rain attacks. A classical mathematical model for describing the fluid motion in unsaturated zone in a porous medium is the Richards equation, a nonlinear degenerate advection-diffusion equation. Two main research directions arose recently for the Richards equation. One is to find exact solutions by finding a specific form of coefficient functions so as to make the equation completely integrable. An exact solution helps to clearly capture the physical mechanism of the phenomena and to pursue controllability [1,2,3]. The other research direction is to seek an approximate solution by numerical methods, for coefficient functions to match with a practical situation. Finite elements, finite difference or finite volumes methods are carried out in [4,5] and the reference therein. The above schemes commonly used fully implicit schemes based on a backward Euler format. Adaptive time stepping is studied in [6,7]. In these studies, some conservative schemes are devised and numerical tests show that they have good stability and some order accuracy, but no theoretical proof is given.

So far, we have only known one paper to prove the stability for the mixed finite element discretization of the Richards equation in [8,9]. In [8], they introduced an implicit mixed element scheme and applied the Kirchhoff transformation to deal with the degeneracy of the Richards equation. The Kirchhoff transformation could be used in the continuous inner product, but we have to take the discrete inner product in the analysis for the difference scheme, so we take a new way which is by adding a perturbation to the coefficient function of parabolic term to overcome the degeneracy.

In [8], the scheme they used is implicit, the stability condition is certainly superior to the explicit scheme. So they do not pay attention to the stability condition in the analysis for the implicit scheme. However, the explicit numerical schemes always are stable only in rigorous restriction for mesh ratio. To improve the stability condition, we introduced an extra term in the difference scheme to relax the time step restriction for improving the stability condition. Also, the theoretical analysis for the implicit numerical schemes and the explicit difference scheme is completely different.

As we know, these early attempts are all implicit schemes based on backward Euler difference scheme and the central difference scheme, although in certain case, the implicit scheme may have to be used to avoid instability. However a strongly nonlinear algebraic system must be solved at each time level, even though these iterative methods [10,11] are used, it still needs huge calculation. Explicit scheme is a good choice to improve the computation efficiency, but the classical explicit scheme cannot be used for the Richards equation due to its degeneracy and the severe time step length restriction.

The main purpose of this work is to provide an efficient explicit numerical scheme for the Richards equation and prove the stability. The key objectives of this work are threefold: First, we add a perturbation to the coefficient function of parabolic term to overcome the degeneracy. Secondly, we introduce a stabilization term with constant coefficient in the difference scheme to relax the stability restriction on the time step. Please note that a similar technique has been used in the simulation of the Cahn–Hilliard equation [12] and the MBE models [13]. The Cahn–Hilliard equation and the MBE models are fourth-order parabolic partial differential equations, so they introduced a second-order stabilization term in the Fourier spectral scheme and finite element scheme respectively. The Richards equation is a second-order equation, so the stabilization term which is added in the explicit difference scheme is completely different. Finally, we prove the stability by induction method and perform some numerical experiments.

The organization of the paper as follows. In Section 2, an explicit difference scheme is given for the Richards equation. In Section 3, we prove the stability of such scheme. In Section 4, some numerical experiments are given. We conclude the paper in Section 5.

## 2. Richards Equation and the Explicit Difference Scheme

The Richards equation could be written in three equivalent forms, with either pressure head h[L] or moisture content $\theta [L^{3}/L^{3}]$ as the dependent variable. We recall that the hydraulic head h+z is partitioned into the pressure head h=p/(ρg) and the gravity head *z*, the vertical coordinate increasing upwards, with the pressure *p* normalized by the gravity force. Here ρ is the mass density of the fluid and *g* is the gravity acceleration. The constitutive relationship between θ=θ(z,t) and h=h(z,t) allows the conversion from one to another. The three forms can be identified as *h*-based, θ-based, and mixed:*h*-based
(1)$$C\left(h\right)\frac{\partial h}{\partial t}-\nabla \cdot K\left(h\right)\nabla h-\frac{\partial K}{\partial z}=0,$$θ-based
(2)$$\frac{\partial \theta }{\partial t}-\nabla \cdot D\left(\theta \right)\nabla \theta -\frac{\partial K}{\partial z}=0,$$mixed
(3)$$\frac{\partial \theta }{\partial t}-\nabla \cdot K\left(h\right)\nabla h-\frac{\partial K}{\partial z}=0,$$
where the real-valued functions C(h)≡dθ/dh, K(h), and D(θ)≡K(θ)/C(θ) respectively denote the specific moisture capacity function [1/L], the unsaturated hydraulic conductivity [L/T] and the unsaturated diffusivity $\left[L^{2}/T\right]$. The coefficient K(h) describes the ease with which water can move through pore spaces, and depends on the intrinsic permeability of the material, degree of saturation, and the density and the viscosity of the fluid. The porous medium is assumed to be isotropic.

We consider the *h*-based form with the datum reported by Haverkamp et al. [14,15] which is used to solve an example of infiltration into soil column.
(4)$$\theta \left(h\right)=\frac{\alpha (\theta_{s}-\theta_{r})}{\alpha +{\left|h\right|}^{\beta }}+\theta_{r},\;K\left(h\right)=K_{s}\frac{A}{A+{\left|h\right|}^{\gamma }},$$
where θ(h) represents the moisture content. $\theta_{r}$ and $\theta_{s}$ are initial and saturated moisture content respectively. Moreover, C(h)=dθ/dh, simple calculations show that
(5)$$C\left(h\right)=\theta^{\prime }\left(h\right)=\frac{\alpha \left(\theta_{s}-\theta_{r}\right)\beta {\left|h\right|}^{\beta -1}}{{\left(\alpha +|h\right|}^{\beta }{)}^{2}},\;K^{\prime }\left(h\right)=\frac{K_{s}A\gamma {\left|h\right|}^{\gamma -1}}{{\left(A+|h\right|}^{\gamma }{)}^{2}}.$$

For the given θ(h) and K(h), we consider the following data [16].
$$\begin{array}{c} \alpha =1.611\times 10^{6},\;\theta_{s}=0.287,\;\theta_{r}=0.075,\;\beta =3.96,\\ K_{s}=0.00944\;\mathrm{cm}/s,\;A=1.175\times 10^{6},\;\gamma =4.74. \end{array}$$
(6)$$h\left(40\;\mathrm{cm},t\right)=h_{top}=-20.7\;\mathrm{cm},\;h\left(0,t\right)=h_{bottom}=-61.5\;\mathrm{cm}.$$

These data provide some real numbers from an example of infiltration into soil column. From the data, we can verify that there are upper bounds for K(h) and $K^{\prime }\left(h\right)$ easily, i.e., $K\left(h\right)\leq K<K_{s},K^{\prime }\left(h\right)\leq K_{1}$ for h∈R. This is to be used in the stability proof.

Let Δz=L/M be the uniform step length, where *M* is a positive integer. We divide the domain of time *T* with *N* segments, let $\tau =T/N,t^{n}=n\tau$ be the uniform time length. Then for a function h(t,z), denote $H_{i}^{n}=h\left(z_{i},t_{n}\right)$, where $z_{i}=m\Delta z,m=0,1,\ldots ,M$, and $\bar{\Omega }=\left\{z_{i}|i=0,1,\ldots M\right\}$, $t_{n}=n\tau ,n=0,1,\ldots ,N$. Let $\lambda =\tau /\Delta z^{2}$ be the mesh ratios.

Define the following difference operators
$$\delta_{t}H_{i}^{n}=\frac{H_{i}^{n+1}-H_{i}^{n}}{\tau },0\leq n\leq N-1,$$
$$\nabla_{h}H_{i}^{n}=\frac{H_{i+1}^{n}-H_{i-1}^{n}}{2\Delta z},\;\;\Delta_{h}H_{i}^{n}=\frac{H_{i+1}^{n}-2H_{i}^{n}+H_{i-1}^{n}}{{\Delta z}^{2}},1\leq i\leq M-1.$$

Now, we introduce the discrete $L^{2}$ inner product as $\langle u,v\rangle =\Delta z\{\frac{1}{2}\left(u_{0}v_{0}+u_{M}v_{M}\right)+\sum_{i=1}^{M-1}u_{i}v_{i}\},$ and the coresponding discrete $L^{2}$ norm is ${∥v∥}_{h}={\langle v,v\rangle }^{\frac{1}{2}}$. Moreover, the discrete $H^{1}$ seminorm ${\left|\cdot \right|}_{1,h}$ and the discrete maximum norm ${\left|\cdot \right|}_{\infty ,h}$ are defined as
$${\left|v\right|}_{1,h}={\left[\Delta z\sum_{i=1}^{M-1}{\left(\nabla_{h}v_{i}\right)}^{2}\right]}^{\frac{1}{2}}{,\;|v|}_{\infty ,h}=\underset{i}{\sup}\left|v_{i}\right|,i=1,\ldots M.$$

A classical first-order explicit difference scheme is
(7)$$C\left(H_{i}^{n}\right)\delta_{t}H_{i}^{n}-\nabla_{h}\left(K\left(H_{i}^{n}\right)\nabla_{h}H_{i}^{n}\right)-K^{\prime }\left(H_{i}^{n}\right)\nabla_{h}H_{i}^{n}=0.$$

For the degeneracy of the Richards equation, a special trick to handle the nonlinear parabolic term is devised. We add a positive constant $\epsilon_{1}$ to $C(H_{i}^{n})$ in (Equation 7) (in fact, $\epsilon_{1}$ can be seen as a small positive bound of $C(H_{i}^{n})$). Then modified first-order explicit scheme is of the form
(8)$$C\left(H_{i}^{n}\right)\delta_{t}H_{i}^{n}+\epsilon_{1}\delta_{t}H_{i}^{n}-\nabla_{h}\left(K\left(H_{i}^{n}\right)\nabla_{h}H_{i}^{n}\right)-K^{\prime }\left(H_{i}^{n}\right)\nabla_{h}H_{i}^{n}=0.$$

With the Richards equation featured by a convection dominated diffusion problems, numerical experiments show that the stability of (Equation 8) is restricted by the mesh ratio, meaning that the scheme is stable only in very tiny time step, as is expected. So we add extra diffusion terms in (Equation 8) to improve the stability condition so that relax the restriction of the time step.
(9)$$C\left(H_{i}^{n}\right)\delta_{t}H_{i}^{n}+\epsilon_{1}\delta_{t}H_{i}^{n}-\epsilon_{2}\left(\Delta_{h}H_{i}^{n+1}-\Delta_{h}H_{i}^{n}\right)-\nabla_{h}\left(K\left(H_{i}^{n}\right)\nabla_{h}H_{i}^{n}\right)-K^{\prime }\left(H_{i}^{n}\right)\nabla_{h}H_{i}^{n}=0,$$
where $\epsilon_{2}$ is a positive constant to be determined so as to improve the stability condition.

**Remark** **1.**
*In [17], for the one-dimensional Richards equation, we also established a linearized semi-implicit finite difference scheme and analyzed the stability. Compared to the scheme of [17], the explicit difference scheme (Equation 9) is to avoid solving a linear algebraic equations at every time step. If we divide the domain of time with N segments, the explicit difference scheme could reduce the computational cost to 1/N of that. So the explicit difference scheme (Equation 9) is more concise and the speed of its numerical simulation is faster.*


## 3. Stability Analysis

**Theorem** **1.**
*The scheme (Equation 9) is stable with $L_{\infty }$ discrete norm, when the time-step length satisfies $\tau <\left(C_{\epsilon }+\epsilon_{1}\right)K_{s}/K_{1}^{2}$, where the $C_{\epsilon }\geq 0$ is the lower bound of the $C(H_{i}^{n})$.*


**Proof.** From (Equation 4), the lower bound of the $K(H_{i}^{n})$ is non-zero for the bounded |h|. Now we assume that there is a constant $k_{\epsilon }>0$ such that $k_{\epsilon }<K\left(H_{i}^{n}\right),i=0,1,\ldots ,M$ for fixed *n*. Taking the inner product of (Equation 9) with $H^{n+1}-H^{n}$ gives
$$I_{1}+I_{2}+I_{3}-I_{4}=0,$$
where $I_{1}$–$I_{4}$ satisfy
$$I_{1}=\langle \left(C\left(H^{n}\right)+\epsilon_{1}\right)\frac{H^{n+1}-H^{n}}{\tau },H^{n+1}-H^{n}\rangle \geq \left(C_{\epsilon }+\frac{\epsilon_{1}}{\tau }\right){∥H^{n+1}-H^{n}∥}^{2},$$
$$I_{2}=-\langle \epsilon_{2}\left(\Delta_{h}H^{n+1}-\Delta_{h}H^{n}\right),H^{n+1}-H^{n}\rangle =\epsilon_{2}{∥\nabla_{h}H^{n+1}-\nabla_{h}H^{n}∥}^{2},$$
$$\begin{array}{cc} I_{3}= & \langle K\left(H^{n}\right)\nabla_{h}H^{n},\nabla_{h}H^{n+1}-\nabla_{h}H^{n}\rangle \\ = & \langle K\left(H^{n}\right)\left(\nabla_{h}H^{n+1}-\nabla_{h}H^{n}\right),\nabla_{h}H^{n}-\nabla_{h}H^{n+1}+\nabla_{h}H^{n+1}\rangle \\ \geq & -K∥\nabla_{h}H^{n+1}-\nabla_{h}H^{n}{∥}^{2}+\frac{K_{\epsilon }}{2}(∥\nabla_{h}H^{n+1}{∥}^{2}-∥\nabla_{h}H^{n}{∥}^{2}), \end{array}$$
and
$$I_{4}=\langle K^{\prime }\left(H^{n}\right)\nabla_{h}H^{n},H^{n+1}-H^{n}\rangle \leq \frac{\epsilon_{1}}{2\tau }∥H^{n+1}-H^{n}{∥}^{2}+\frac{K_{1}^{2}\tau }{2\epsilon_{1}}{∥\nabla_{h}H^{n}∥}^{2}.$$Summing up, we obtain
$$\begin{array}{c} \left(C_{\epsilon }+\frac{\epsilon_{1}}{2\tau }\right)∥H^{n+1}-H^{n}{∥}^{2}+\left(\epsilon_{2}-K\right){∥\nabla_{h}H^{n+1}-\nabla_{h}H^{n}∥}^{2}\\ +\frac{K_{s}}{2}(∥\nabla_{h}H^{n+1}{∥}^{2}-∥\nabla_{h}H^{n}{∥}^{2})\leq \frac{K_{1}^{2}\tau }{2\epsilon_{1}}{∥\nabla_{h}H^{n}∥}^{2}, \end{array}$$
for $\epsilon_{2}\geq K$. Simple calculation shows that
(10)$$∥\nabla_{h}H^{n+1}{∥}^{2}\leq \left(1+\frac{K_{1}^{2}\tau }{\epsilon_{1}K_{s}}\right){∥\nabla_{h}H^{n}∥}^{2},$$
which implies that there exists a positive constant $c_{1}$, being independent of Δz and τ, such that $∥\nabla_{h}H^{n+1}{∥}^{2}\leq c_{1}$.We are now prepared to prove the stability with respect to the discrete $L^{2}$ norm. Taking the inner product of (Equation 9) with $H^{n+1}$, we have
$$E_{1}+E_{2}+F_{1}\leq E_{3}+F_{2}+E_{5},$$
where
$$E_{1}=\langle \left(C\left(H^{n}\right)+\epsilon_{1}\right)\frac{H^{n+1}-H^{n}}{\tau },H^{n+1}\rangle \geq \frac{C_{\epsilon }+\epsilon_{1}}{2\tau }(∥H^{n+1}{∥}^{2}-∥H^{n}{∥}^{2}),$$
$$E_{2}=-\langle \epsilon_{2}\Delta_{h}H^{n+1},H^{n+1}\rangle =\epsilon_{2}{∥\nabla_{h}H^{n+1}∥}^{2},$$
$$E_{3}=-\langle \epsilon_{2}\Delta_{h}H^{n},H^{n+1}\rangle \leq \frac{\epsilon_{2}}{2}(∥\nabla_{h}H^{n+1}{∥}^{2}+∥\nabla_{h}H^{n}{∥}^{2}),$$
$$\begin{array}{cc} E_{4}= & \langle K\left(H^{n}\right)\nabla_{h}H^{n},\nabla_{h}H^{n+1}\rangle =\langle K\left(H^{n}\right)\nabla_{h}H^{n},\nabla_{h}H^{n+1}-\nabla_{h}H^{n}+\nabla_{h}H^{n}\rangle ,\\ = & \underset{F_{1}}{\underbrace{\langle K\left(H^{n}\right)\nabla_{h}H^{n},\nabla_{h}H^{n}\rangle }}+\underset{F2}{\underbrace{\langle K\left(H^{n}\right)\nabla_{h}H^{n},\nabla_{h}H^{n+1}-\nabla_{h}H^{n}\rangle }}, \end{array}$$
$$E_{5}=\langle K^{\prime }\left(H^{n}\right)\nabla_{h}H^{n},H^{n+1}\rangle \leq \frac{K_{\epsilon }}{2}∥\nabla_{h}H^{n}{∥}^{2}+\frac{K_{1}^{2}}{2K_{\epsilon }}{∥H^{n+1}∥}^{2}.$$Applying Young’s inequality, we obtain
$$\begin{array}{c} F_{1}\geq K_{\epsilon }∥\nabla_{h}H^{n}{∥}^{2},\;\;F_{2}\leq \frac{K}{2}(∥\nabla_{h}H^{n+1}{∥}^{2}-∥\nabla_{h}H^{n}{∥}^{2}). \end{array}$$Eventually, we obtain
$$\left(\frac{C_{\epsilon }+\epsilon_{1}}{2\tau }-\frac{K_{1}^{2}}{2K_{\epsilon }}\right)∥H^{n+1}{∥}^{2}+\left(\frac{K_{\epsilon }}{2}-\frac{\epsilon_{2}}{2}+\frac{K}{2}\right)∥\nabla_{h}H^{n}{∥}^{2}+\left(\frac{\epsilon_{2}}{2}-\frac{K}{2}\right)∥\nabla_{h}H^{n+1}{∥}^{2}\leq \frac{C_{\epsilon }+\epsilon_{1}}{2\tau }{∥H^{n}∥}^{2}.$$We conclude that for $\tau <\left(C_{\epsilon }+\epsilon_{1}\right)K_{s}/K_{1}^{2}$, there is a positive constant $c_{2}$ which is independent of Δz and τ such that
(11)$$∥H^{n+1}{∥}^{2}\leq c_{2}.$$Combining the above two results (Equation 11) and (Equation 10), using Sobolev’s embedding inequality, we can get $|H^{n+1}{|}_{\infty ,h}$ is bounded when n→∞. It implies the inductive hypothesis holds, completing the proof. □

Similar estimation techniques were used in other useful applications [18].

**Remark** **2.**
*We exploit the constant $\epsilon_{2}$ to improve the stability. If $\epsilon_{2}>K$, the scheme (Equation 9) is stable. Please note that $K<K_{s}$ and $K_{s}=0.00944cm/s$ makes it possible to take $\epsilon_{2}$ sufficiently small in case of excessive errors.*


**Remark** **3.**
*To make sure the numerical scheme (Equation 9) is convergent, the time step should be chosen to satisfy $\tau <\left(C_{\epsilon }+\epsilon_{1}\right)K_{s}/K_{1}^{2}$. If $C_{\epsilon }=0$, we have to take a non-zero perturbation $\epsilon_{1}$ to ensure the stability of the scheme. Because of $C_{\epsilon }>0$, we are allowed to take $\epsilon_{1}=0$ in numerical experiments.*


## 4. Numerical Experiments

In this section, we illustrate the numerical stability by a numerical experiment of the infiltration process based on a generalized *h*-based Richards equation. Since it is difficult to obtain the exact solution of this model, to verify the theoretical results, we take the following non-homogeneous model,
$$\left\{\begin{array}{c} C\left(h\right)\frac{\partial h}{\partial t}-\nabla \cdot \left(K(h)\nabla h\right)-\frac{\partial K}{\partial z}=g\left(z,t\right),\\ h(z,0)=-1.02z-20.7, \end{array}\right.$$
where the boundary conditions remain unchanged as (Equation 6). If we suppose an exact solution h=−1.02−20.7+t(z−40)/(4T), a simple calculation shows that
$$\begin{array}{cc} g(z,t)= & \frac{5808800331328389z\left(z-40\right){\left(\frac{51z}{50}-\frac{tz(z-40)}{4T}+\frac{207}{10}\right)}^{74/25}}{17179869184T{\left[{\left(\frac{51z}{50}-\frac{tz(z-40)}{4T}+\frac{207}{10}\right)}^{99/25}+1611000\right]}^{2}}\\ \; & -\frac{5546t}{{\left(\frac{51z}{50}-\frac{tz(z-40)}{4T}+\frac{207}{10}\right)}^{237/50}+1175000}\\ \; & -\frac{3613000706430075\left(\frac{t(z-20)}{2T}-\frac{51}{50}\right){\left(\frac{51z}{50}-\frac{tz(z-40)}{4T}+\frac{207}{10}\right)}^{187/50}}{68719476736{\left[{\left(\frac{51z}{50}-\frac{tz(z-40)}{4T}+\frac{207}{10}\right)}^{237/50}+1175000\right]}^{2}}. \end{array}$$

By using the scheme (Equation 9), we show, in Figure 1, the variation trend of the pressure head with depth for the time interval from 10 s to 360 s, with the choice of $\epsilon_{1}=0$ and $\epsilon_{2}=0$.

In [14], the authors used an implicit numerical scheme, and took a large time step, for instance, 10 s, 30 s and 120 s, to save the computational workload. In this paper, we take the time step as small as $10^{-3}$ s. Correspondingly the space steps that they used are much larger than ours. The different scheme and the large grid gap may bring some but tolerable discrepancy.

We take T=100 s and M=200 to test the stability of the scheme with different time steps and different choices of $\epsilon_{2}$. The results are listed in Table 1. It is evident that the improvement in the stability by use of the extra terms is significant. Moreover, in Table 2, we set $\epsilon_{1}=0,\epsilon_{2}=0,M=200$ and T=1 s, and confirm that the expected order of convergence is obtained.

Figure 2 shows the linear relationship between $\epsilon_{1}$ and errors.

## 5. Conclusions

In this work, a stable explicit scheme for the Richards equation was developed and analyzed. We proposed techniques to avoid the degeneracy of the Richards equation and improve the stability condition of the finite difference scheme. A numerical example is provided to verify our theoretical analysis. Demonstration of the numerical stability over a long time, along with the error estimate as shown by Figure 2, is indicative of the physical stability of a typical solution of the Richards equation; infinitesimal perturbations to the solution do not grow. A rigorous mathematical analysis of the stability of the traveling-wave solution and its relevance to the numerical stability call for an independent investigation.

Compared to implicit numerical schemes and linearized numerical schemes, stable explicit numerical schemes improve the calculation efficiency. This paper is a first step toward the explicit difference schemes for the Richards equation, we only analysis the stability of such a scheme. It is our ongoing work to extend other high order accuracy explicit difference schemes and estimate the errors.

## Figures and Tables

**Figure 1 entropy-22-00352-f001:**
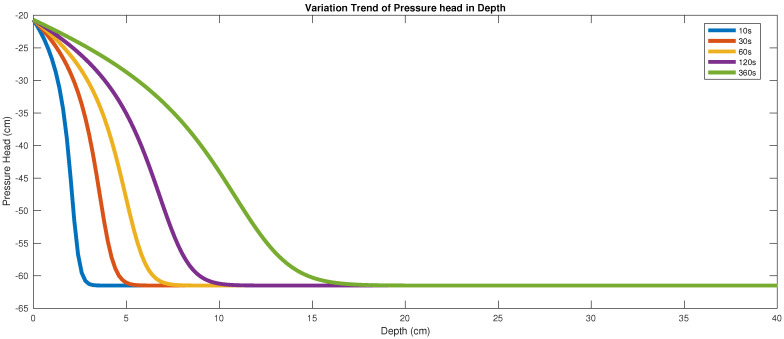
Variation of the pressure head with depth.

**Figure 2 entropy-22-00352-f002:**
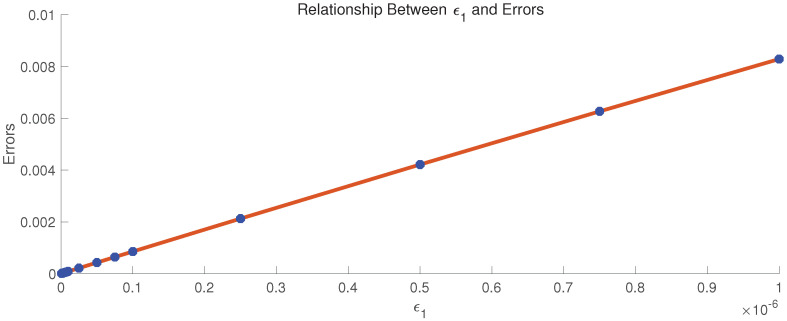
Relationship between $\epsilon_{1}$ and errors.

**Table 1 entropy-22-00352-t001:** Stability comparison with different $\epsilon_{2}$ and τ.

		τ	τ=0.4	τ=0.2	τ=0.1	τ=0.05	τ=0.025	τ=0.0125
	Accuracy							
$\epsilon_{2}$								
$\epsilon_{2}=0$			Unstable	Unstable	Unstable	Unstable	$5.60\times 10^{-4}$	$3.26\times 10^{-5}$
$\epsilon_{2}=0.0001$			Unstable	Unstable	Unstable	Unstable	$1.38\times 10^{-4}$	$3.75\times 10^{-5}$
$\epsilon_{2}=0.0005$			Unstable	$1.88\times 10^{-1}$	$5.47\times 10^{-2}$	$5.00\times 10^{-3}$	$3.78\times 10^{-4}$	$1.90\times 10^{-4}$
$\epsilon_{2}=0.001$			$9.10\times 10^{-2}$	$1.44\times 10^{-2}$	$4.00\times 10^{-3}$	$1.50\times 10^{-3}$	$7.54\times 10^{-4}$	$3.79\times 10^{-4}$

**Table 2 entropy-22-00352-t002:** Accuracy.

N	1000	2000	4000	8000	16,000
${\left|h-H\right|}_{\infty ,h}$	$1.90\times 10^{-3}$	$9.65\times 10^{-4}$	$4.82\times 10^{-4}$	$2.41\times 10^{-4}$	$1.21\times 10^{-4}$
Ratio	Non	0.98	1.00	1.00	0.99

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
