# Peer review of "Stability Analysis of the Explicit Difference Scheme for Richards Equation"

_entropy, 2020, doi:10.3390/e22030352_

Round 1
Reviewer 1 Report
This paper studies a stable explicit scheme for the Richards equation. The authors proposed techniques to avoid the degeneracy of the Richards equation and improve the stability condition of the finite difference scheme. A numerical example is provided to verify the theoretical analysis. As far as I can see, the results seem to be new. The topic is certainly worth investigation. However, the presentation can be improved. There are some technical issues that need to be clarified. The detailed comments are as follows. I will give my final recommendation based on the revision.
(1) According to my search in researcher database, the first author seems put the first/last name in the reverse order. If so, please rectify.
(2) It is mentioned that there are few theoretical results for the numerical schemes of the Richards equation up to now in Introduction. Therefore, the novelty should be better highlighted in Introduction. For a paper to be published in Entropy, a certain degree of novelty is essential.
(3) Related to the above comment, in line 33-34, it is mentioned a similar technique has been used in the simulation of the Cahn-Hilliard equation etc. It would be good to highlight the difference/difficulty of using this in the current context.
(4) Eq(1) and the h-based form should be explained. Please keep in mind that Entropy is an interdisciplinary journal with readers from a wide range of backgrounds.
(5) In Eq(4), is there any special consideration for the data used? It seems that the authors applied the same data as in Ref.[16]. The necessity should be explained.
(6) In Remark 1, the mentioned faster convergence should be justified.
(7) The last inequality on page 3 is not easy to follow. I do not see why it holds. An explanation is needed.
(8) At the end of the proof of Theorem 1, it may be remarked that similar estimation techniques have been used in other useful applications. e.g., 'Fixed-time group consensus for multi-agent systems with non-linear dynamics and uncertainties'.
(9) The choice of g(z,t) is interesting. Is there any special reason to choose this form?
(10) Tables are good, but the information is very limited. It would suggest adding figures rather than tables. Moreover,
(11) The conclusion section is very short. It would be beneficial to mention some future directions and open problems.
(12) Some journals have different forms in your reference list. Please make them consistent. For example, in Ref. [2] 'Adv Water Resource' but in Ref. [9] 'Adv Water Res'. These little parts all contribute to the quality of the paper in the end.
Author Response
Response to Referee's Report We are very grateful to the referee for careful reading and valuablesuggestion of our manuscript. We have revised some details as pointed
by the referee. For this manuscript, the details are as follows.
THE FIRST REVIEWER: 1) According to my search in researcher database, the first author seems put the first/last name in the reverse
order.If so,please rectify. {\bf Answer:} I am sorry to make this mistake for my neglect. I rectified the name as Fengnan Liu in the revised manuscript. 2) It is mentioned that there are few theoretical results for the numerical schemes of the Richards equation up to
now in introduction.Therefor, the novelty should be better highlighted in introduction.For a paper to be published in
Entropy, a certain degree of novelty is essential. {\bf Answer:} Thanks for your suggestion. As we know, all the numerical schemes for the Richards equation up to
now are implicit schemes.We introduced a linearized finite difference based on backward Euler format in [5], this scheme improved the calculation efficiency. To further improve the calculation efficiency, we introduced the explicit difference scheme in this manuscript.The theoretical analysis for the implicit numerical schemes and the explicit difference scheme is completely different.
Based on your suggestion, we add following contents in introduction to describe our novelty: `` So far, we have just known one paper prove the stability for the mixed finite element discretization of Richards equation in [1,15]. In reference [15], they introduced an implicit mixed element scheme and applied the Kirchhoff transformation to deal with the degeneracy of Richards equation.The Kirchhoff transformation could be used in the
continuous inner product, but we have to take the discrete inner product in the analysis for the difference scheme, so we take a new way which is by adding a perturbation to the coefficient function of parabolic term to overcome the degeneracy. In [15], the scheme they used is implicit, the stability condition is certainly superior to the explicit scheme. So they do not pay attention to the stability condition in the analysis for the implicit scheme. But the explicit numerical schemes always are stable only under rigorous restriction for mesh ratio. In order to improve the stability condition, we have introduced an extra term in the difference scheme to relax the time step restriction for improving the stability condition. Also, the theoretical analysis for the implicit numerical schemes and the explicit difference scheme is completely different.'' 3) Related to the above comment, in line 33-34, it is mentioned a similar technique has been used in the simulation
of the Cahn-Hiliard equation etc. It would be good to highlight the difference/difficulty of using this in the current
context. {\bf Answer:} Thanks for your suggestion. ``The Cahn-Hilliard equation and the MBE models are fourth-order
parabolic partial differential equations, so they introduced a second-order stabilization term in the Fourier spectral
scheme and finite element scheme respectively. Richards equation is a second-order equation, so the stabilization
term which is added in the explicit difference scheme is completely different.'' We highlight the difference as the
above contents in introduction. 4) Eq(1) and the h-based form should be explained. Please keep in mind that Entropy is an interdisciplinary journal
with readers from a wide range of backgrounds. {\bf Answer:} We are sorry for that we did not give a clear explanation for Eq(1). We add the explanation for the Richards equation in Section 2 as:
``with either the pressure head $h[L]$ or the moisture content $\theta[L^3/L^3]$ as the dependent variable. The constitutive relationship between $\theta=\theta(t,z)$ and $h=h(t,z)$ allows the conversion from one to another.
Suppose that the porous medium is isotropic.
The three forms can be identified as $h$-based, $
\theta$-based, and mixed:
\begin{itemize}
\item $h$-based
\begin{equation}
\label{1-1}
C(h)\frac{\partial h}{\partial t}-\nabla\cdot K(h)\nabla h-\frac{\partial K}{\partial z}=0,
\end{equation}
\item $\theta$-based
\begin{equation}
\label{1-2}
\frac{\partial\theta}{\partial t}-\nabla\cdot D(\theta)\nabla\theta-\frac{\partial K}{\partial z}=0,
\end{equation}
\item mixed
\begin{equation}\label{1-3}
\frac{\partial\theta}{\partial t}-\nabla\cdot K(h)\nabla h-\frac{\partial K}{\partial z}=0,
\end{equation}
\end{itemize}
where $z$ denotes the vertical coordinate and is assumed to be positive in the upward direction.
The real-valued functions $C(h)\equiv d\theta/d h$, $K(h)$ and
$D(\theta)\equiv K(\theta)/C(\theta)$ respectively denote the specific moisture capacity function $[1/L]$, unsaturated hydraulic conductivity $[L/T]$, and unsaturated diffusivity $[L^2/T]$. The function $K(h)$ describes the ease with which water can move through pore spaces, and depends on the intrinsic permeability of the material, the degree of saturation, and the density and the viscosity of the fluid." 5) In Eq. (4), is there any special consideration for the data used? It seems that the authors applied the same data as in Ref. [16]. The necessity should be explained. {\bf Answer:} Thanks for your question. Yes, the data we used in the manuscript is realistic data which is given to illustrate the infiltration into
soil column [6]. We have not been able to find other realistic data up to now. Following your suggestion, we modified the expression about the data as:
``These data provide realistic values from an example of infiltration into soil column.’’ 6) In Remark 1, the mentioned faster convergence should be justified. {\bf Answer:} Thanks for your suggestion. In [12], for the one-dimensional Richards equation, we also established a linearized semi-implicit finite difference scheme and analyzed the stability. Comparing with the scheme of [12], the explicit difference scheme (\ref{s3}) is to avoid solving a linear algebraic equations at every time step. If we divide the domain of time with $N$ segments, the explicit difference scheme could reduce the computational cost to $1/N$ of that. So the explicit difference scheme (2.9) is more concise and the speed of its numerical simulation is faster.''
Remark 1 is amended as above on your suggestion. 7) The last inequality on page 3 is not easy to follow. I do not see why it holds. An explanation is needed. {\bf Answer:} Thank you. We rewrite this part in the revised manuscript. Taking the inner product with $H^{n+1}$, we have
$$
E_1+E_2+E_4\leq E_3 +E_5,
$$
where
\begin{equation}\nonumber
E_1=\langle(C(H^{n})+\epsilon_1)\frac{H^{n+1}-H^{n}}{\tau},H^{n+1}\rangle\geq\frac{C_{\epsilon}+\epsilon_1}{2\tau}(\|H^{n+1}\|^2-\|H^{n}\|^2),
\end{equation}
\begin{equation}\nonumber
E_2=-\langle\epsilon_2\Delta_hH^{n+1},H^{n+1}\rangle =\epsilon_2 \|\nabla_hH^{n+1}\|^2,
\end{equation}\begin{equation}\nonumber
E_3=-\langle\epsilon_2\Delta_hH^{n},H^{n+1}\rangle \leq \frac{\epsilon_2}{2}( \|\nabla_hH^{n+1}\|^2+\|\nabla_hH^{n}\|^2),
\end{equation}
\begin{eqnarray}
E_4&=&\langle K(H^{n})\nabla_hH^{n},\nabla_hH^{n+1}\rangle=\langle K(H^{n})\nabla_hH^{n},\nabla_hH^{n+1}-\nabla_hH^{n}+\nabla_hH^{n}\rangle,
\nonumber\\
&=& \underbrace{\langle K(H^{n})\nabla_hH^{n},\nabla_hH^{n}\rangle}_{F_1}+\underbrace{\langle K(H^{n})\nabla_hH^{n},\nabla_hH^{n+1}-\nabla_hH^{n}\rangle}_{F2},\nonumber\end{eqnarray}
\begin{equation}
E_5=\langle K'(H^n)\nabla_hH^n,H^{n+1}\rangle \leq \frac{K_{\epsilon}}{2}\|\nabla_h H^{n}\|^2+\frac{K_1^2}{2K_{\epsilon}}\|H^{n+1}\|^2.\nonumber
\end{equation}
Applying Young's inequality, we obtain
\[
\begin{array}{l}
F_1\geq K_{\epsilon} \|\nabla_h H^n\|^2,\quad \quad -F_2\leq \frac{K}{2}( \|\nabla_h H^{n+1}\|^2- \|\nabla_h H^n\|^2).
\end{array}
\]
Eventually, we obtain
\begin{eqnarray}&&
\left(\frac{C_{\epsilon}+\epsilon_1}{2\tau}-\frac{K_1^2}{2K_{\epsilon}}\right)
\|H^{n+1}\|^2
+\left(\frac{K_{\epsilon}}{2}-\frac{\epsilon_2}{2}+\frac{K}{2} \right)
\|\nabla_h H^n\|^2
\nonumber\\&&
+\left(\frac{\epsilon_2}{2}-\frac{K}{2} \right)\|\nabla_h H^{n+1}\|^2
\leq \frac{C_{\epsilon}+\epsilon_1}{2\tau}\|H^{n}\|^2.
\nonumber
\end{eqnarray} 8) At the end of the proof of Theorem 1, it may be remarked that similar estimation techniques have been used in
other useful applications. E.g.,` Fixed-time group consensus for multi-agent systems with non-linear dynamics and
uncertainties'. {\bf Answer:} Thanks for your suggestion. We have added a sentence as:
``The similar estimation techniques have been used
in other useful applications [16]'' at the end of the proof of Theorem 1. 9) The choice of $g(z,t)$ is interesting. Is there any special reason to choose this form? {\bf Answer:} Thanks for your question. Yes, the reason is for precisely testing the calculation results. The exact solution of the Richards
equation which is considered in this manuscript cannot be found, so the calculation results only be built between
numerical solutions, that will cause the distortion. So we suppose an exact solution for a non-homogenous model,
a simple calculation shows $g(z,t)$ as this form. We are sorry for our unclear explanation for it, and we redefine the
relationship between the exact solution and the $g(z,t)$ in Section 4. 10) Tables are good, but the information is very limited. It would suggest adding figures rather than
tables.
{\bf Answer:} Thanks for your suggestion. We are sorry for this, but we have only gotten the stability for the explicit difference scheme in this manuscript. The theoretical analysis for the convergence and errors is our next aim. So more figures about the convergence and errors will be given in our next paper. But thanks for your advice again. 11) The conclusion section is very short. It would be beneficial to mention some future directions and open problems. {\bf Answer:} Thanks for your question. This is excellent advice as the information of the conclusion is limited. We have added the following sentences in Conclusion to mention our future work:
``Comparing with implicit numerical schemes and linearized numerical schemes, stable explicit numerical schemes improve the calculation efficiency. This paper is a first step toward the explicit difference schemes for the Richards equation, with making an analysis only of the stability of such a scheme. It is our ongoing work to extend other high order accuracy explicit difference schemes and estimate the errors.'' 12) Some journals have different forms in your reference list. Please make them consistent. For example, in Ref.[2]
'Adv Water Resource' but in Ref. [9] 'Adv Water Res.' These little parts all contribute to the quality of the paper in
the end. {\bf Answer:} Thanks for your kindly advice. We took the unified form in the reference list. Thank you very much! REFERENCES: F. Liu, Y. Fukumoto, X. Zhao, A linearized finite difference scheme for the Richards equation under
variable-flux boundary conditions, J. Sci. Comput. (2020) to appear. F. Radu, I. S. Pop, P. Knabner, Order of convergence estimates for an Euler implicit, mixed finite element
discretization of Richards equation, SIAM J. Numer. Anal., \textbf{42} (2004),1452-1478. M. A. Celia, E. T. Bouloutas, A general mass-conservative numerical solution for the unsaturated flow equation,
Water Resource Research, \textbf{26} (1990),1483-1496. Y. Shang, Fixed-time group consensus for multi-agent systems with non-linear dynamics and uncertainties, IET control Theory Appl., \textbf{12} (2018), 395-404.

Reviewer 2 Report
This paper presents a study on the stability analysis of the explicit difference scheme for the Richards equation. A perturbation to the functional coefficient of the parabolic term is introduced to avoid the degeneracy of the Richards equation. The stability was proved by the induction method. A numerical example is carried out. It is an interesting paper. However, the manuscript is not of sufficient quality as well as novelty to be published in a journal. The main reasons are as follows.
No significant results or findings are addressed in the abstract. As there are many studies in the finite difference method for solving nonlinear problems, the novelties and the originality offered in this study are not significant. Comprehensive literature reviews in nonlinear problems using the finite difference method should be provided. Only one numerical example is carried out. To demonstrate the quality and applicability, more numerical examples should be conducted. Besides, it is not clear why the g(z, t) in the numerical example is with the specific term. In Equation (4), the authors should give the proper unit of the pressure head. In addition, the value of the pressure head should consider the possible maximum and minimum values with the knowledge of the infiltration of water into the ground. Overall, this manuscript may be more welcome in the pure mathematical journal and put more energy on the mathematical issues. The Richards equation is a highly nonlinear equation governed by nonlinear physical relationships which are described using soil water characteristic curves. The topic of this manuscript is the Richards equation. Unfortunately, the contents of this manuscript are divergent from the Richards equation and the only numerical example may be not realistic in physics.Author Response
Response to Referee's Report
We are very grateful to the referee for careful reading and valuablesuggestion of our manuscript. We have revised some details as pointed
by the referee. For this manuscript, the details are as follows. THE SECOND REVIEWER: Thanks for your kind advice. We are sorry for our unclear explanation for $g(z,t)$, and we redefine
the relationship between the exact solution and the $g(z,t)$ in Section 4.
As regards the value of the pressure head, we have provided the results in [5]. In this manuscript, we focus on the stability of the explicit difference scheme, and have proposed two new ways to overcome the degeneracy of the model and to relax the stability condition of the explicit difference scheme respectively.
Comparing with implicit numerical schemes and linearized numerical schemes, a stable explicit numerical scheme dramatically improve the calculation efficiency.
From the view of our work, it is important work for solving Richards equation by proposing a stable explicit difference scheme and provide the
stability of such scheme. And we think such techniques in the stability analysis could be used for other high order
accuracy explicit difference schemes.
We pay attention to the numerical analysis in this manuscript, so we submit it to the Entropy special issue on
``Applications of Nonlinear Diffusion Equation'', with our expectation that this topic might well fit with it.
Thank you very much for your suggestion again.
Reviewer 3 Report
The authors present an explicit scheme for numerically solving the Richards equation. Stability analysis is performed. The presented results are restricted to 1D. A numerical example is shown to sustain the theoretical results. The paper is suitable for the journal. The topic is interesting.
Nevertheless, there are a few things to be improved before I can recommend the publication of the manuscript.
The proof of the stability is not clear to me. It does not look like the authors obtain a contraction. For example in (8), one concludes that the left hand side is bounded, but actually in the boundary will appear a term like (1 + a)^n which obviously will go to infinity when n goes to infinity. The same problem I see it before eq. (9). I do not see how the authors conclude that (9) results when tau is less... These points should be clarified or corrected.2. What is the reason to restrict the analysis to 1D?
3. Why the stability condition is not depending on the spatial mesh size? This looks strange to me.
4. The authors should mention also other parametrizations, not only Haverkamp. Especially they should mention the non-Lipschitz case (C() becomes unbounded), treated in
https://link.springer.com/chapter/10.1007/978-3-319-96415-7_3
5. The type of regulations the authors performed was also done earlier in
https://www.sciencedirect.com/science/article/pii/S037704270301001X
This should be mentioned as well.
Author Response
Response to Referee's Report
We are very grateful to the referee for careful reading and valuable
suggestion of our manuscript. We have revised some details as pointed
by the referee. For this manuscript, the details are as follows.
THE THIRD REVIEWER: 1) The proof of the stability is not clear to me. It does not look like the authors obtain a contraction. For example in
(8), one concludes that the left hand side is bounded, but actually in the boundary will appear a term like $(1+a)^n$
which obviously will go to infinity when $n$ goes to infinity. The same problem I see it before eq.(9). I do not see
how the authors conclude that (9) results when tau is less. These points should be clarified or corrected. {\bf Answer:} Thanks for your suggestion. As we known, $n\leq N=1/\tau$, the equation $\lim\limits_{N\rightarrow
\infty}(1+\tau)^{N}=e$ is a common one in numerical analysis. For the equation (8), $\|\nabla_hH^{n+1}\|^2 \leq
(1+\frac{K_1^2\tau}{\epsilon_1 K_s}) \|\nabla_hH^{n}\|^2$, when $n\rightarrow \infty$ we have $(1+
\frac{K_1^2\tau}{\epsilon_1K_s})^n\leq\lim\limits_{N\rightarrow\infty}(1+\frac{K_1^2\tau}{\epsilon_1K_s})^N
=e^{\frac{K_1^2}{\epsilon_1K_{\epsilon}}}$.
We transform the inequality (9) into $\|H^{n+1}\|^2\leq (1+\frac{K_1^2\tau}{(C_\epsilon+\epsilon_1)K_{\epsilon}-
K_1^2\tau})\|H^{n}\|^2$,
then we can get $\|H^{n}\|^2\leq e^{\frac{K_1^2}{(C_{\epsilon}+\epsilon_1)K_{\epsilon}}}$ when $n\rightarrow\infty$. 2) What is the reason to restrict the analysis to 1D? {\bf Answer:} Thanks for your question. Certainly 2D and 3D models would be more relevant to natural phenomena. The Richards equation is a degenerate nonlinear advection-diffusion equation which models the fluid motion in unsaturated zone in a porous medium. For the study of underground water, the main factors to affect the fluid motion in unsaturated zone are the time and the vertical coordinate. Hence, the starting point should be a 1D model only related to the time $t$ and the vertical coordinate $z$. Moreover, from the viewpoint of mathematics and physics, because of its degeneracy and nonlinearity, it is difficult to study the properties of its solution, even for the 1D model. Hence, many papers (including our manuscript) restrict the analysis to the 1D Richards equation. In this manuscript, we have tried to establish a valid useful numerical scheme to analyze the 1D Richards equation. 3) Why the stability condition is not depending on the spatial mesh size? This looks strange to me. {\bf Answer:} The explicit difference scheme for the Richards equation is depending on the mesh ratio, so it is depending on the spatial mesh size definitely. The technique which is proposed in the manuscript improves the rigorous restriction on the mesh ratio by using the $\epsilon_2$. The numerical results are shown in Table 1.
In Table 1, the mesh ratio adjustment is achieved by adjusting the $\tau$. 4) The authors should mention also other parametrizations, not only Haverkamp. Especially they should mention the non-Lipschitz case becomes unbounded treated in \\
$https://link.springer.com/chapter/10.1007/978-3- 319-96415-7-3$. {\bf Answer:} Thanks for your suggestion. We list this paper in the reference [17]. 5) The type of regulations the authors performed was also done earlier in \\
$https://www.sciencedirect.com/science/article/pii/S0 37704270301001X$. This should be mentioned as well. {\bf Answer:} Thanks for your suggestion. We list this lecture notes in the reference [11]. REFERENCES: J. W. Both, K. Kumar, J. M. Nordbotten, I. S. Pop, F. A. Radu, (2019) Iterative Linearisation Schemes for
Doubly Degenerate Parabolic Equations. In: F. Radu, K. Kumar, I. Berre, J. Nordbotten, I. Pop, (eds) Numerical Mathematics and Advanced Applications, ENUMATH 2017. Lecture Notes in Computational
Science and Engineering, vol 126. Springer, Cham. I. S. Pop, F. Radu, P. Knabner, Mixed finite elements for the Richards' equation: linearization procedure, J. Comput. Appl. Math., \textbf{168} (2004), 365-373.

Round 2
Reviewer 1 Report
The authors have made good effort to improve the paper. All my comments in the previous round of review have been addressed. Therefore, I am glad to recommend the paper for publication.
Author Response
Thank you very much for your kind advice!
Reviewer 2 Report
Though the authors have revised the manuscript, it is still not of sufficient contents as well as novelty to be published in a journal. Comparing the revised manuscript of version 2 to version 1, the revisions are very limited, mainly on the literature reviews.
In Equation (6), the authors should give the proper unit of the pressure head. The value of the pressure head should consider the possible maximum and minimum values with the knowledge of the infiltration of water into the ground. Though the authors mentioned the pressure heads in Equation (6) were taken from reference [5], the points are the proper values of the pressure heads with the knowledge of the infiltration of water into the ground should be considered in this manuscript. In the paper of Celia et al. [5], they investigated different scenarios for the pressure heads from -20.7 cm~-61.5 cm to -75 cm~-1000 cm to demonstrate the convergence of the finite difference and the finite element methods. Finally, the contents of the manuscript are not sufficient as a journal paper. Additional numerical experiments should be conducted. As the reviewer, I would like to maintain the “Rejection” suggestion.Author Response
THE SECOND REVIEWER: Thanks for your kind advice. We have added Figure 1, a figure to express evolution of the pressure head by a numerical experiment with use of one of the data borrowed from Celia's paper [6]. In equation (2.6), we have given the unit `cm' to the pressure head $h$. To explain this unit, we have inserted, at the beginning of Section 2, the following sentence:``We recall that the hydraulic head $h+z$ is partitioned into the pressure head $h=p/(\rho g)$ and the gravity head $z$, the vertical coordinate increasing upwards, with the pressure $p$ normalized by the gravity force. Here $\rho$ is the mass density of the fluid and $g$ is the gravity acceleration." In section 4, we have amended typo of the first equation:
``$h(x,0)$"\ $\to$\ ``$h(z,0)$" About the other data which is mentioned in your comments are not only different in the condition and initial values, but also have significant differences in the coefficient functions. Actually, the two data sets correspond to two separate models. Then, the base and the analytic method would be different. Beside, Celia's paper focuses on the numerical simulation and our work on numerical analysis backed by mathematical foundation. In this manuscript, we put emphasis on the stability of the explicit difference scheme, and propose two new ways to overcome the degeneracy of the model and to relax the stability condition of the explicit difference scheme respectively. The first technique for overcoming the degeneracy can be applicable to other degenerate models. The second technique may serve as a new way for improving stability condition of explicit difference scheme for other convection-diffusion models. Also, since the previous references on the numerical approximation of the Richards equation commonly used fully implicit schemes based on a backward Euler format, perhaps we are the first to compute the Richards equation by explicit numerical scheme, and to provide mathematical support for the scheme. We believe that this work is worth recording. For the purpose of testing our numerical scheme, we think it reasonable to select a typical example which is close to a realistic model with its parameter values fitting with a measurement. This paper conducts mathematical analysis of a new numerical scheme for integrating the degenerate nonlinear diffusion equation. Accordingly, we have submitted it to the Entropy special issue on
``Applications of Nonlinear Diffusion Equation'', with our expectation that this topic might well fit with it.
Thank you very much for your suggestion again.

Reviewer 3 Report
The authors answered satisfactory all my comments.
Author Response

(The authors gave the same response as above.)
